# Temporal Dynamics of Host Use by *Drosophila suzukii* in California’s San Joaquin Valley: Implications for Area-Wide Pest Management

**DOI:** 10.3390/insects10070206

**Published:** 2019-07-15

**Authors:** Xingeng Wang, Gülay Kaçar, Kent M. Daane

**Affiliations:** 1Beneficial Insects Introduction Research Unit, Agricultural Research Service, United States Department of Agriculture, Newark, DE 19713, USA; 2Department of Environmental Science, Policy, and Management, University of California Berkeley, Berkeley, CA 94720-3114, USA; 3Faculty of Agriculture and Natural Sciences, Bolu Abant Izzet Baysal University, 14280 Bolu, Turkey

**Keywords:** spotted wing drosophila, alternative host, fruit crop, non-crop host, host suitability

## Abstract

A major challenge to the area-wide management of *Drosophila suzukii* is understanding the fly’s host use and temporal dynamics, which may dictate local movement patterns. We determined *D. suzukii*’s seasonal host use in California’s San Joaquin Valley by sampling common crop and non-crop fruits in a temporal sequence of fruit ripening. We then evaluated the suitability of selected fruits as hosts. *Drosophila suzukii* emerged from both intact and damaged cherries during the cooler, early season period. Fly density remained low through the hot spring–summer period and re-surged as temperatures lowered in fall when the fly did not cause damage to intact peach, nectarine, plum, pear, grape, pomegranate, apple, persimmon and citrus (in order of ripening) but did emerge from the damaged fruits of these crops. The fly also emerged from two ornamental fruits (loquats and cactus) but was not found on wild plum and two endemic wild fruits (buckthorn and bitter berry). *Drosophila suzukii* completed development (egg to adult) on cactus, mandarin carpel, pomegranate seed, wild plum and buckthorn at survival rates similar to cherry (51.2–68.8%), whereas it had a lower survival rate on bitter cherry (33.2%), table grape (31.5%), raisin grape (26.5%), and wine grape (4.5%). The high acidity levels of grapes negatively affected the fly’s fitness. Among 10 cherry cultivars, survival rate was not affected by sugar content, but it decreased with increasing egg density per gram of fruit. Results suggest that in California’s San Joaquin Valley, the early season crops are most vulnerable, summer fruits ripen during a period of low pest pressure, and late season fruits, when damaged, serve to sustain *D. suzukii*’s populations in this region.

## 1. Introduction

Native to East Asia, *Drosophila suzukii* (Matsumura) (Diptera: Drosophilidae) has been a major invasive pest of soft-and thin-skinned fruits since it was first detected in 2008 in North America and Europe [1,2,3] and has been found recently in South America [4,5]. *Drosophila suzukii* is highly polyphagous, being able to oviposit and/or reproduce in various cultivated and wild fruits [6,7,8,9,10,11]. Its fast development and high reproductive potential can lead to explosive population increases [12,13,14] and significant economic losses to crops [15,16]. Though various management strategies, including behavioral, biological, chemical and cultural approaches, have been implemented to suppress *D. suzukii* populations and reduce crop damage [17,18,19,20,21,22,23,24], current control programs rely heavily on insecticides that target adult flies in commercial crops [25,26]. Because non-crop habitats can act as a reservoir for the fly’s reinvasion into treated crops [27,28,29], area-wide Integrated Pest Management (IPM) strategies that reduce population densities at the landscape level need to be developed for such a highly mobile and polyphagous pest.

To develop area-wide programs, it is critical to understand how *D. suzukii* populations persist and disperse in the landscape as the season progresses. Many environmental factors, such as local climatic and landscape traits, may trigger the dispersal of *D. suzukii* populations to escape resource-poor habitats or unfavorable weather conditions [30,31,32,33]. Landscape composition surrounding cultivated crops, such as forests and shrub vegetation, could act as sinks, sources, shelters or overwinter sites for the fly populations [34,35]. For this reason, the availability of alternative hosts could play an important role in sustaining fly populations and dictating their local movement patterns when favorable hosts are not available. Researchers have provided a better understanding of local *D. suzukii* population dynamics [36,37,38,39,40,41]. Still, there are gaps that limit our understanding of the relative importance of different hosts for *D. suzukii* within some geographical regions. For example, the seasonal periods of host utilization and the importance of non-crop hosts within the agricultural landscape need to be understood to develop area-wide programs. In this framework, this study aimed to illustrate the temporal dynamics of host use by *D. suzukii* in California’s San Joaquin Valley, one of the world’s major fruit growing regions.

*Drosophila suzukii* was first detected in California (and North America) when it was found infesting strawberries and cranberries in Santa Cruz County in 2008 [42]. Since then, damaging populations have been recorded from cherries, cranberries, mulberries, raspberries and strawberries, mainly in the coastal or northern California fruit growing regions with relatively mild summer [42,43,44,45]. In comparison, California’s interior San Joaquin Valley has hotter summers and colder winters [46], and while *D. suzukii* is collected in cherry, citrus, fig, grape, kiwi, mulberry, nectarine, peach, persimmon, plum and pomegranate as well as in non-crop habitats surrounding the orchards [39,47,48], reported crop damage has been mainly on cherries [48]. Adult fly captures show two main periods of activity—spring and fall—and low captures in winter (except for evergreen riparian areas) and summer [39,47,48]. The number of captured flies was positively related between pairs of sampled sites based on their proximity, but it was negatively related to differences in fruit ripening periods among crops, suggesting that fly populations might move among crop and/or non-crop habitats during the year [39].

Though adult flies are captured in various orchard crops, it is not clear whether these fruits are vulnerable and serve as hosts. For example, the potential impact of *D. suzukii* on wine grapes in Italy was discussed by Ioriatti et al. [49], who observed *D. suzukii* oviposition in soft-skinned berries, and, in Japan, some grape cultivars were reported as hosts for *D. suzukii* [7]. In Oregon, Lee et al. [9] found that *D. suzukii* was able to successfully oviposit in some wine grape cultivars but that offspring survival was low (<10%), whereas other studies observed no or low levels of infestation of intact grapes in the field or laboratory [50,51,52,53,54,55,56,57]. Some of the initial work in Japan reported that *D. suzukii* emerged only from fallen and damaged apple, apricot, loquat, peach, pear, persimmon and plum [7], but Sasaki and Sato [58] reported that healthy peach fruit can be infested. However, in California, Stewart et al. [59] reported that intact peach fruit are unlikely hosts. No doubt, many fruits with hard or hairy skin can be colonized if wounds are available to allow flies to oviposit in the pulp [4,59,60,61].

In this study, we document the temporal patterns of host use by *D. suzukii* in California’s San Joaquin valley by sampling intact and damaged fruits of various crop and no-crop plants throughout the fruiting season. We evaluated the suitability of key fruits, including several unreported ornamental and wild host fruits as hosts for the fly, particularly focusing on the host status of grapes—considered to be a non-preferred host—and cherry—considered to be a preferred host. Wine grapes can contain uniquely high levels of organic acids (with ideal pH range of 3.3–3.7) that are important for producing wines less susceptible to microbial and oxidative damage and with more vibrant color [62]. The levels of acidity decrease as fruit are ripening, but they remain high throughout the ripening process [63,64]. For this reason, we also examined the impact of tartaric acid concentrations on the fly’s fitness. For cherries, we examined the effects of cultivar and fruit size on the fly’s performance. We additionally monitored adult fly populations at different elevations—from the Valley floor east to the foothills and Sierra mountains—to determine if the fly is active at higher elevations during the hot summer when the fly populations were extremely low in the Valley’s agricultural areas [39]. We discuss the implications of this information for area-wide management in the San Joaquin Valley.

## 2. Materials and Methods

### 2.1. Fruit Sampling

A total of 17 common fruits were sampled in a temporal sequence of fruit ripening, including twelve important crops (cherry, peach, nectarine, plum, pear, grape, fig, pomegranate, apple, persimmon, mandarin orange and naval orange), three ornamentals (loquat, wild plum and cactus), and two wild host plants (bitter cherry and buckthorn) (Table 1). Samples were taken from 2013 to 2015 at the University of California’s Kearney Agricultural Research and Extension Center, near Parlier, California (Fresno County) and near Brentwood, California (Contra Costa County). Ornamental fruits were also collected in riparian areas surrounding agricultural crops near Bentwood. Bitter cherry, *Prunus emarginata* (Douglas ex Hooker) Eaton (Rosaceae), and the Cascara buckthorn, *Frangula purshiana* (DC.) are endemic to western North America; these fruits were collected at higher elevations 1683 m) near Shaver Lake, California (Fresno County) (Table 1). For all species, both intact fruit (without obvious damage) and damaged fruit (with open wound, split or cracks) were collected as available, as the fruit were at a susceptible ripening stage for *D. suzukii* oviposition. A total of 30–50 fruit were collected when at a susceptible ripening stage for each species, although the number of intact ornamental and wild fruits varied depending on the availability.

Collected fruits were placed individually (large fruits) or in groups of 10–50 (small fruits) in deli cups (11 × 11 cm) and held under controlled conditions (23 ± 2 °C, 14 L:10 D, 40–70% Relative Humidity (RH) at the University of California’s Kearney Agricultural Research and Extension Center (Kearney). Deli cups were covered with fine organdy cloth and fitted with a raised metal grid (2 cm high) on the bottom to suppress mold growth. A piece of tissue paper was placed underneath the fruit to absorb any liquid accumulation. Emerged flies were collected every 2–3 d, frozen, and then identified as either *D. suzukii* or other drosophilids. Only those flies that emerged within 2 weeks following field collection were counted to exclude the possibility of second-generation flies.

### 2.2. Laboratory Tests

#### 2.2.1. Insects

All laboratory studies were conducted under controlled conditions, as described above (Section 2.1). A laboratory colony of *D. suzukii* was established from field collections of infested cherries at Kearney. The fly larvae were maintained on a standard cornmeal-based artificial diet using methods described by Dalton et al. [66], and adult flies were held in Bug Dorm2 cages (BioQuip Products Inc., Rancho Dominguez, CA, USA) supplied with a 10% honey–water solution and petri dishes containing standard cornmeal medium sprinkled with brewer’s yeast for feeding and oviposition. Field-collected *D. suzukii* were introduced into the colony yearly to maintain the vigor of the colony. All tests used 1–2-week-old adult female flies that had been housed with males since emergence (and therefore assumed to be mated).

#### 2.2.2. Host Suitability

The suitability of 10 different fruits ovipositional and/or reproductive hosts for *D. suzukii* were evaluated in no choice tests: Pomegranate (*Punica granatum*) ‘*cv.* Wonderful,’ mandarin orange (*Citrus reticulata*) ‘*cv.* Seedless Kishu,’ grapes (*Vitis vinifera*) (table ‘*cv.* Crimson Seedless,’ raisin ‘*cv.* Thompson Seedless,’ and wine ‘*cv.* Cabernet Sauvignon’ cultivars), wild plum (*Prunus subcordata*), prickly cactus (*Optunia streptacantha*), Cascara buckthorn (*F. purshiana*), and bitter cherry (*P. emarginata*). As a companion, the cherry (*Prunus avium*) ‘*cv.* Bing’ was also included. To test the effect of different cherry cultivars on the fly’s performance, 10 cultivars were selected to include fruit with different colors, shapes, and sizes. All fruits were obtained during fruit sampling described previously; crop fruits were collected from unsprayed fields and carefully examined under microscope to ensure there was no natural infestation by *D. suzukii* prior to the test. Because we pre-determined that *D. suzukii* is unable to lay eggs into intact pomegranate, orange, and cactus fruit, individual pomegranate seeds, citrus carpels, and halved cactus fruit were used. Additionally, wild plum was punctured using 1.0 mm diameter needles to facilitate fly oviposition. The intact and whole fruit was used as a test unit.

Tests for each fruit species consisted of 22–53 replicates. Tested fruits (or fruit unit) were weighed prior to each test, and Brix (sugar content) was measured from a sub-sample using a refractometer (ATC-1E Brix 0–32%, ATAGO USA Inc., Bellevue, WA, USA) and for the fruit firmness (surface penetration force) using a penetrometer with a 1 mm test tip (L-500 g, 5 g/Div., AMETEK Inc., Berwyn, PA, USA) (a value of 100 g mm^−1^ indicates that 100 g of force is needed to penetrate the 1 mm diameter section of fruit surface [66]). Replicates were a single fruit or fruit unit exposed to a female *D. suzukii* for 24 h in a ventilated, acrylic cage (8 × 11 × 14 cm), with 10% honey water provided via cotton wicks in a 50 mL vial as food for the fly. After the female fly was removed and the number of eggs in the fruit was counted, fruits were kept in cages until the emergence of the flies. 

#### 2.2.3. Fruit Size Preference 

Ripe ‘*cv.* Bing’ cherries were used for a choice test to determine the fly’s preference between large and small fruit and the effects of fruit size on offspring survival. The test was conducted in the acrylic cages. Each fruit was weighed, and one small (4–5 g) and one large (8–9 g) fruit were placed inside a test cage and exposed to one female fly for 24 h, with 10% honey water provided via cotton wicks in 50 mL vial. Following exposure, the number of eggs on each fruit was counted, and the small and large fruit were placed in different cages until the emergence of adult flies. The test had 35 replicates. 

#### 2.2.4. Egg Density Effect

To determine the effect of *D. suzukii* egg density on the percentage of eggs successfully developed to adult flies, large fruit (8–9 g) of ripe ‘*cv.* Bing’ cherries were individually exposed to single female flies, in the cages, for different periods of time (2 h to 2 d) to manipulate the number of *D. suzukii* eggs laid per fruit. A total of 93 fruits were exposed with a density ranging from 1 to 43 eggs per fruit (the fruit with the extremely high density of 43 eggs was excluded from the analysis). All exposed fruit were weighed and kept in the cages until the emergence of adult flies.

#### 2.2.5. Citrus Test

To determine if *D. suzukii* can oviposit within and develop from damaged or rotting navel oranges (‘*cv.* Thomson Improved’), a single adult female *D. suzukii* was exposed to a whole fresh fruit, halved fresh fruit, rotting whole fruit, or halved rotting fruit for 24 h in the acrylic cage. To simulate the natural decay process of a fallen orange, fresh oranges were placed individually on wet sandy soil in deli cups (11 × 11 cm) until the fruit started to rot. The halved fruit were allowed the same amount of time as the whole fruit but were cut into halves just prior to the test. On average, rotted fruit had 42.3 ± 7.3% of their surface covered by mold growth. Following exposure, the numbers of eggs laid were counted, and the fruit was then held in the cage until the emergence of adult flies. Each treatment started with 25 replicates; however, a few replicates were discarded because of contamination by other drosophilids that likely occurred during the regular examination for the decay status of the fruit. A sub-sample of 10 fruit was measured to determine the Brix levels of fresh and rotting fruits. 

#### 2.2.6. Grape Acidity Effect

To determine the possible effect of tartaric acid on *D. suzukii* survival and development, seven different concentrations of tartaric acid (0, 2, 4, 6, 8, 10 or 12 g/L) were mixed with a standard artificial diet. The powdered tartaric acid was purchased from a wine and beer brewing store in Fresno, CA, USA, and mixed with the diet just before the diet solidified. The content of tartaric acid in grapes can vary depending on cultivar, ripeness, and environmental conditions; for example, Kliewer et al. [63] reported a tartaric acid content ranging from 3.7 to 13.2 g/L in different cultivars and from 3.4 to 9.2 g/L in early- vs. late-harvested cultivars. The doses used here covered these reported ranges. Each treatment had 20–22 replicates, and each replicate started with 10 *D. suzukii* eggs (<24 h) from the laboratory culture that were placed in drosophila vials over the diet. The number of developed adults was recorded. A sub-sample of 25 pupae from each treatment was measured for pupal length (*l*) and width (*w*), and the volume (*V*) of each pupa was estimated based on the formula (*V* = 4/3π (*l*/2) (*w*/2)^2^) [67].

### 2.3. Monitoring of Adult Populations at Different Elevations 

Apple cider vinegar traps were used to monitor fly populations at four different elevations from the Valley’s low agricultural areas to the Sierra Nevada: Kearney (106 m), lower foothills (272 m), higher foothills (525 m) and Sierra mountains (1683 m). Traps at Kearney were placed in a mixed stone fruit orchard; traps at the three higher elevations were along Highway 168, with the foothill sites in residential yards with fruit trees (citrus, grape, apple or plum) and the Sierra site at the forest’s edge in bitter cherry bushes. Three traps were placed at each location, approximately 200 m apart. Collection methods were similar to Wang et al. [39]. Briefly, traps were constructed of plastic containers filled with apple cider vinegar (Great Value Apple Cider Vinegar^®^, Wal-Mart Stores, Inc., Bentonville, AR, USA) and a small amount of Bon-Ami Free and Clear^®^ unscented soap (Bon-Ami Company, Kansas City, MO, USA) to serve as a surfactant. Traps were hung on tree branches at head-height and then checked and replaced weekly from June to November 2017. Captured arthropods were placed into 95% ethanol in small glass bottle and later examined under a dissecting microscope to count the number of *D. suzukii*. 

### 2.4. Data Analysis

Counts of emerged adult *D. suzukii* were based on total fruit samples. Laboratory data on fruit size preference, citrus test and grape acidity effect were presented as treatment means (± SE), and treatment effects were compared with the Analysis of Variance (ANOVA).

For host suitability tests, since fruit varied in weight, the percentage of *D. suzukii* eggs that successfully developed to adults was calculated based on eggs per gram fruit to standardize the comparison among different treatments. The egg density effect on the percentage of eggs developed to adults in cherry was analyzed using linear regression. The percentage of eggs that successfully developed to adults on different fruit species or different cherry cultivars were subject to further analysis of generalized linear model (GLM) with binomial distribution and log-link function by considering the effect of both fruit species or cultivar and egg density per gram fruit, as well as the interactions of these two factors. To separate the means among different treatments, the percentage data were also arcsine transformed as needed to normalize the variance and analyzed using ANOVA. All analyses were performed using JMP V13 (SAS 2013, Cary, NC, USA). 

## 3. Results

### 3.1. Fruit Sampling

*Drosophila suzukii* emerged from both intact and damaged cherries but from only damaged apples, citrus, figs, loquats, nectarine, peaches, persimmons, plums, pears, cactus, and cracked pomegranates (Table 2). No *D. suzukii* emerged from grapes, wild plums, buckthorn and bitter berries (Table 2). The numbers of emerged *D. suzukii* from different fruits varied, with high numbers of flies emerging from damaged pears in Brentwood and figs in Fresno. Cherries were most seriously infested by *D. suzukii*; for example, a mean of 1.2 ± 0.2 *D. suzukii* emerged per fruit (*n* = 723) during May and June in 2013, with a peak infestation on 25 May as high as 2.0 ± 0.2 flies per fruit (*n* = 50). Detailed information on the field sampling results are provided in Appendix A. 

### 3.2. Host Suitability

Tested fruit species differed in firmness (F_5,109_ = 154.6, *p* < 0.001), Brix level (F_7,147_ = 325.2, *p* < 0.001), size (per fruit unit) (F_9,375_ = 925.9, *p* < 0.001), eggs pre fruit (F_9,375_ = 28.2, *p* < 0.001), or eggs per gram fruit (F_9,375_ = 27.6, *p* < 0.001) that followed a similar pattern as eggs per fruit (Table 3). Firmness varied from 45.6 to 181.54 g mm^−1^ for all fruits in which oviposition occurred. The fly failed to oviposit in wild plum, which had a firmness of 295.4 g mm^−1^ (this fruit was thus punctured). The percentages of eggs that developed to adults on cactus, mandarin carpel, pomegranate seed, wild plum, and buckthorn were similar to that on cherry (51.2–68.8%), but the success rate of development from egg to adult was lower on bitter cherry (33.2%), table grape (31.5%), raisin grape (26.5%), and wine grape (4.5%) (Table 3). GLM analyses showed that the success rate was affected by the fruit species (χ^2^ = 58.7, df = 9, *p* < 0.001) and the number of eggs per gram fruit (χ^2^ = 10.8, df = 1, *p* < 0.001) but not the interaction of these two factors (χ^2^ = 11.5, df = 9, *p* = 0.243).

Among the 10 tested cherry cultivars, fruits differed in size (F_9,91_ = 50.8, *p* < 0.001), firmness (F_9,140_ = 48.5, *p* < 0.001), Brix level (F_9,90_ = 33.4, *p* < 0.001), and the number of eggs laid per gram fruit unit (F_9,240_ = 7.1, *p* < 0.001) (Table 4), but the number of eggs per fruit was not different (F_9,240_ = 0.8, *p* = 0.627) (Table 4). The percentages of eggs that developed to adults differed among cherry cultivars (F_9,240_ = 3.1, *p* = 0.001) (Table 4), but the difference did not result from the cultivar but rather the egg density per gram of fruit (cultivar: χ^2^ = 9.5, df = 9, *p* = 0.389; eggs per gram of fruit: χ^2^ = 25.3, df = 1, *p* < 0.001; cultivar x eggs/gram fruit: χ^2^ = 8.1, df = 9, *p* = 0.522). Note that Cultivar 6 was the smallest fruit. Consequently, the egg density per gram fruit on Cultivar 6 was extremely high, resulting in a low percentage of eggs developed to adults. There was no relationship between the mean percentage of *D. suzukii* eggs that successfully developed to adults and the mean Brix level of the cultivar (F_1,9_ = 0.7, *p* = 0.419, r^2^ = 0.083) or between the fruit firmness and the number of eggs laid per gram fruit (F_1,9_ = 0.5, *p* = 0.518, r^2^ = 0.054).

### 3.3. Fruit Size Preference 

In choice tests with two different size fruit (4.4 ± 0.1 g and 8.7± 0.1 g for small and large fruit, F_1,68_ = 1296.0, *p* < 0.001), although adult females laid more eggs in larger than smaller fruit (F_1,68_ = 6.1, *p* = 0.016) (Figure 1A), the number of eggs per gram fruit was similar (F_1,68_ = 0.1, *p* = 0.730) (Figure 1B). Thus, although the fly preferred large over small fruit in terms of the number of total eggs allocated to differently sized fruit, the percentage of eggs that developed to adults (y) was not affected by fruit size (F_1,68_ = 0.001, *p* = 0.973) (Figure 1C), but the percentage of *D. suzukii* eggs that successfully developed to adult flies decreased with increasing egg density per gram fruit (Figure 2).

### 3.4. Orange Test 

Naval orange condition affected the number of eggs laid (F_3,96_ = 24.6, *p* < 0.001) (Figure 3A). No eggs were oviposited into intact fruit, although three eggs were found on the surface of two intact fruit (i.e., unsuccessful oviposition where an egg was not inserted vertically into the fruit flesh, but the entire egg was placed horizontally over the fruit surface). 

The average surface penetration of the fresh fruit was 321.1 ± 16.9 g mm^−1^ (*n* =10), which was about 2–3 times higher than that of the tested cherry fruit (Table 3). When the orange fruit was rotting, the flies laid some eggs into the gelatinous rind of the rotting fruit. After the fruit was halved, the fly readily oviposited into the fruit flesh and rotting fruit; the mean numbers of eggs laid in halved fresh fruit was similar to that in halved rotting fruit (Figure 3A). The percentage of eggs that successfully developed to adults from the halved fresh fruit (39.7 ± 8.3%) or halved rotting fruit (67.9 ± 9.8%) was not significantly different (F_1,36_ = 2.9, *p* = 0.099) (Figure 3B). At the time of oviposition, the average Brix levels of the fresh (10.7 ± 0.2) and rotting fruit (10.5 ± 0.2) were similar (F_1,18_ = 0.3, *p* = 0.567).

### 3.5. Grape Acidity

Tartaric acid concentration negatively affected developmental time (F_6,839_ = 261.8, *p* < 0.001; Figure 4A) and offspring survival from egg to adult (F_6,135_ = 12.3, *p* < 0.001; Figure 4B). The offspring survival rate decreased while developmental time increased with increasing tartaric acid concentration. The body size of developed pupae was similar across different tartaric acid concentrations but smaller than the control treatment (F_6,168_ = 3.9, *p* < 0.001).

### 3.6. Monitoring of Adult Populations at Different Elevations

The mean number of weekly captures of adult *D. suzukii* varied among elevations (Figure 5). At the lowest elevation where most cultivated crops were planted, there were two peaks (spring and fall), and the spring peak (104.7 ± 51.1) was higher than the fall peak (17.0 ± 7.5), but there were no captures during the hottest period of summer and dramatically lower numbers of captured flies throughout other periods of the summer. At the two middle elevations (272 and 525 m), the patterns of captured flies were similar, with a fall peak and a lower number of captured flies throughout the spring and summer. At the highest site (1683 m), no flies were captured during the spring and fall, but a few flies were captured during the hottest period of summer. 

## 4. Discussion

Adult *D. suzukii* were captured in different cultivated and natural habitats in the San Joaquin Valley [39,47,48]; however, we show that in California’s San Joaquin Valley, only the early season crops, such as cherries, appear to be vulnerable to large fly populations, whereas other sampled crops (apple, citrus, fig, grape, nectarine, peach, pear, persimmon, plum, and pomegranate), ornamental plants (cactus, loquat and wild plum), and wild fruits (bitter berry and buckthorn) were either not in a susceptible fruit stage during periods of high density or were not as aggressively attacked because of fruit condition (e.g., fuzzy, thick peach skin). Cherry fruit ripening occurs during a spring peak of fly abundance, likely from overwintered adults or pupae [39,68]. In contrast, fly populations declined rapidly dropped in late June and remained low during the summer and early fall, likely due to raising temperature [39,47,48], when many other fruits were ripening (e.g., figs, nectarines, peaches, pears, plums, figs, and pomegranates). Though the fly population increased in fall, there was not obvious damage to most late-season fruiting crops (apple, citrus, grapes and persimmon). Some of these fruits are known to be utilized by *D. suzukii* as breeding hosts when the fruits were damaged [28,59,69,70]. Among them, cactus, pomegranate seeds, and mandarin carpels are reported for the first time as breeding hosts for *D. suzukii*. Though the number of emerged flies from those fruits was generally low, these alternative fruits may play an important role in the resurgence of the late fall populations and sustaining *D. suzukii* populations late in the season when the fly’s favorable hosts are no longer available in the field. Recoveries of *D. suzukii* from cactus and loquats in the riparian areas adjacent to cherry orchards in Brentwood also suggest the possibility that these ornamental fruits may serve as alternative hosts for *D. suzukii*. Captures of adult *D. suzukii* in the foothills and high mountains were low, and the sampling of both buckthorn and bitter berry did not yield any flies. This suggests that these wild plants have not yet been used by the fly as alternative hosts, even though laboratory trials found them suitable for fly development.

As expected, host suitability was affected by fruit species. Fruit species differ in chemical properties (e.g., sugar content and titratable acidity) and physical traits (e.g., firmness of pulp and toughness of skin) and these factors could affect fly oviposition and offspring development [6,9,59,69,70,71,72,73,74]. In general, oviposition preference decreased with increasing skin hardness, and larval performance increased with increasing in sugar content. Lee et al. [6] evaluated a range of varieties of blackberries, blueberries, cherries, raspberries, and strawberries and found all fruits susceptible to *D. suzukii* once they ripen, but the susceptibility varied with variety and generally increased with the increasing Brix levels. While not attempting to address the oviposition preference or possible effects of other chemical attributes directly, our study nonetheless provides evidence that performance on cactus, mandarin carpel, pomegranate, wild plum and buckthorn (all with a wide range of sugar content) was similar to that on cherry, with the exception of bitter cherry and grapes.

A separate analysis with 10 different cherry cultivars did not yield a significant effect of the Brix on the percentage of eggs that successfully developed to adults, although we could not rule out the possibility that Brix and other chemical properties may affect other fitness parameters of the developed flies (e.g., body size and developmental time) [69]. Many of the differences in chemical traits among different fruits could be attributed to geographic location and differences in environmental and cultivation conditions (soil, climate properties, etc.) rather than inherent varietal properties, such as in cherries [75]. In the current study, chemical differences were controlled to some extent, as cherry cultivars used were grown in the same plot with the same fertilization and irrigation regimes. The physical properties (size or surface firmness) of different cultivars did not seem to affect the fly’s oviposition. The percentage of eggs that developed to adults decreased with the increasing egg density per gram of fruit probably due to intra-specific competition [72], and this was further confirmed by manipulating the egg density and using the same ‘Bing’ cherry cultivar as the tested host. Females preferred larger fruit for oviposition, which is consistent with the density-dependent survival as the large fruit support higher numbers of fly larvae per fruit. It is well known that many fruit flies employ a variety of fruit characters to assess host quality and tend to be more attracted to larger fruits [67,76]. Female *D. suzukii* appears to be able to assess host quality based on fruit size, and this behavior would likely increase foraging efficiency per unit time.

Though we recovered very low numbers of *D. suzukii* from damaged citrus fruits, our laboratory study showed the fly can oviposit into and develop from freshly damaged or rotting navel oranges. Kaçar et al. [68] showed that *D. suzukii* overwinter in citrus, surviving 3–4 months when fresh oranges were provided as adult food or ovipositional medium, and field-emerged adults from soil-buried pupae could produce and oviposit viable eggs on halved mandarin fruit. Thus, citrus fruit likely play an important role as reservoirs in sustaining the fly populations during San Joaquin Valley winter seasons, and in the spring, those populations may migrate into early season crops, such as cherries [48]. 

We did not observe grape infestation in our field collections, and our laboratory trials showed a low survival rate of *D. suzukii* offspring on grapes when compared to other fruits (Table 3). The oviposition susceptibility and offspring survival could vary among varieties or cultivars due to the variations in skin hardness and chemical properties [49,50,51,52,53,54,55,56,57,69]. For example, Ioriatti et al. [53] demonstrated that oviposition increased consistently as the skin hardness of the grape decreased. Chemical properties, such as sugar content and acidity levels, may play a role in host susceptibility. In the current study, we found that although table grapes had a tougher skin than raisin or wine grape cultivars tested (Table 3), females were able to lay eggs into all three types of grapes, often through the fruit surface (43%) or near the petiole (57%). The sugar levels of all tested grapes were either equal to or considerably higher than other fruits tested. We also found that tartaric acid concentration negatively affected the fly’s developmental performance. Still, about 20% eggs successfully developed to adults in the diet mixed with the highest tartaric acid, whereas only 4.5% eggs developed from the wine grape cultivar tested. It is thus possible that other unknown chemical traits might also affect larval performance. Overall, our results are consistent with other reported studies that grapes are not good reproductive hosts for *D. suzukii* [49,50,51,52,53,54,55,56,57,69].

## 5. Conclusions

California’s San Joaquin Valley is one of the world’s most important fruit production regions, with a diverse agricultural landscape that can consist of a mosaic of cultivated and unmanaged host fruit crops. Such diverse landscapes result in the inevitable presence of *D. suzukii* populations that represent a difficult challenge for the management of this polyphagous pest. We showed that only the early seasonal fruits, such as cherries, seem to be at greatest risk to *D. suzukii* (although we did not test blueberries, raspberries or strawberries, which are increasingly being planted in the San Joaquin Valley) Many of other later seasonal fruits are not as vulnerable to this pest, because either their intact skin reduces oviposition, they ripen during a period of low *D. suzukii* abundance, or their flesh has chemical attributes that retard survival. However, some of these alternative hosts—such as citrus and damaged, unharvested stone fruit—may act as shelters for overwintering populations and provide sources for early populations moving into the more susceptible crops. Consequently, area-wide management strategies may need to consider (1) fruit sanitation to lessen overwintering populations, (2) suppressing fall and winter populations by releasing natural enemies, and (3) reducing pest pressure in susceptible crops through ‘border-sprays’ and/or ‘mass trapping’ to kill adults before they move into the vulnerable crop. Alternative and sustainable area-wide management strategies such as biological control are highly desirable to naturally regulate the fly population, especially in uncultivated habitats [77,78,79]. An understanding of the temporal and spatial dynamics of the fly populations would be of aid in the optimal timing of the future release of biological control agents to reduce the source populations in the agricultural landscape [78,79].

## Figures and Tables

**Figure 1 insects-10-00206-f001:**
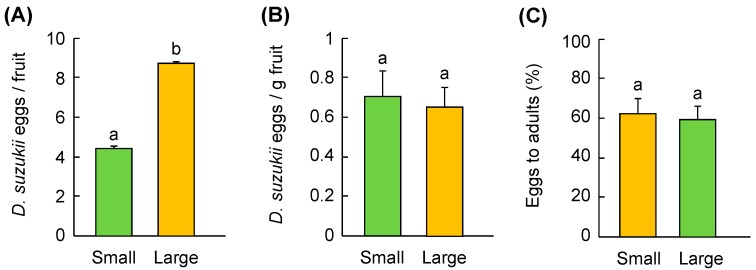
Preference by adult female *D. suzukii* for two different sized cherry fruit: (**A**) Number of eggs laid per fruit, (**B**) number of eggs per gram of fruit, and (**C**) percentage eggs developed to adults. Bars refer to mean ± SE, and different letters over the bars indicate significant difference (One-way ANOVA, *p* < 0.05).

**Figure 2 insects-10-00206-f002:**
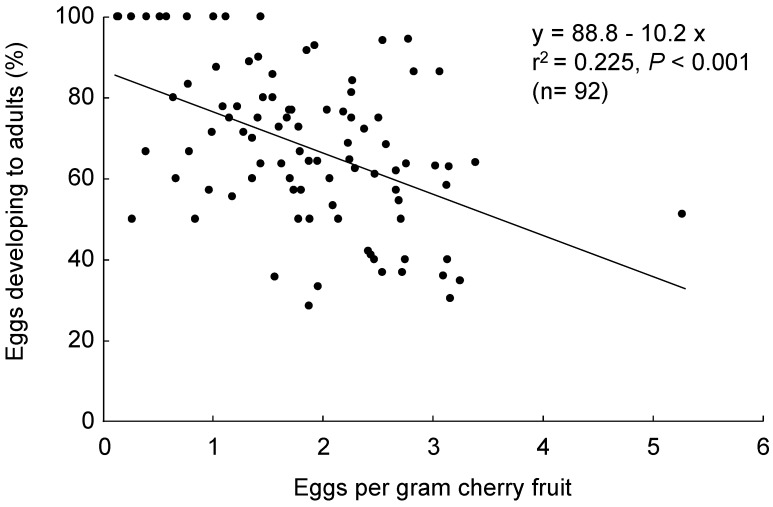
Percentage of *D. suzukii* eggs successfully developed to adult flies decreased with increasing egg density per gram of fruit.

**Figure 3 insects-10-00206-f003:**
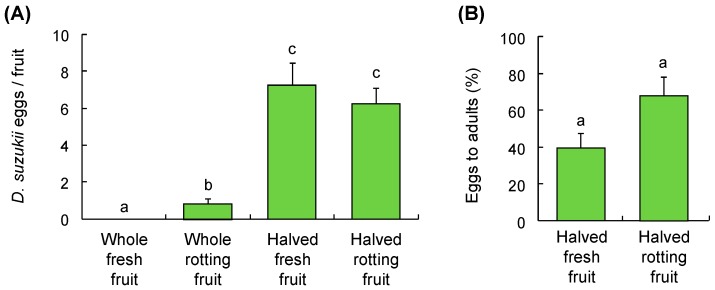
(**A**) Oviposition by *D. suzukii* on harvested citrus with different conditions and (**B**) percentage of eggs that developed to adults. Bars refer to mean ± SE, and different letters over the bars indicate significant difference (One-way ANOVA and Tukey’s HSD, *p* < 0.05).

**Figure 4 insects-10-00206-f004:**
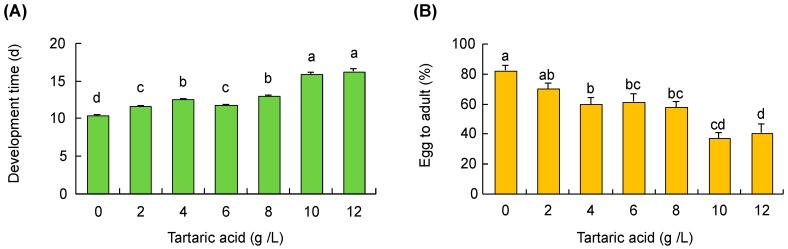
Effects of tartaric acid on (**A**) developmental time and (**B**) the offspring survival (eggs developed to adults) of *D. suzukii*. Bars refer to mean ± SE, and different letters over the bars indicate significant difference (One-way ANOVA and Tukey’s HSD, *p* < 0.05).

**Figure 5 insects-10-00206-f005:**
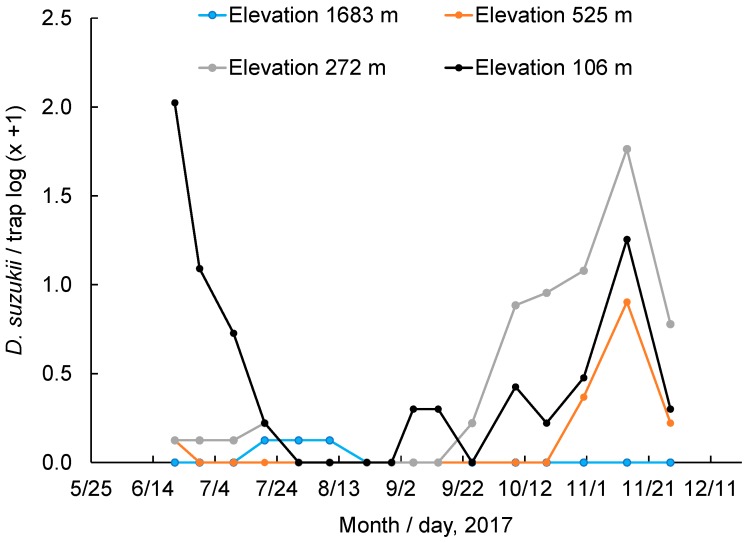
Weekly mean *D. suzukii* trap captures in cedar vinegar traps at different elevations in the San Joaquin valley and its nearby foothills and mountain forest areas.

**Table 1 insects-10-00206-t001:** Periods of fruit ripening and seasonal sampling for *D. suzukii* in crop and non-crop fruits in the San Joaquin Valley, CA, USA.

Family	Host Species	May	Jun	Jul	Aug	Sept	Oct	Nov	Dec to Apr
Rosaceae	Cherry (*Prunus avium*)								
Rosaceae	Peach (*Prunus persica*)								
Rosaceae	Nectarine (*Prunus armeniaca*)								
Rosaceae	Plum (*Prunus domestica*)								
Rosaceae	Pear (*Pyrus domestica*)								
Vitaceae	Raisin grape (*Vitis vinifera*)								
Moraceae	Fig (*Ficus carica*)								
Lythraceae	Pomegranate (*Punica granatum*)								
Rosaceae	Apple (*Pyrus malus*)								
Ebenaceae	Persimmon (*Diospyros kaki*)								
Rutaceae	Naval orange (*Citrus C.* × *sinensis*)								
Rutaceae	Mandarin orange (*Citrus reticulata*)								
Rosaceae	Loquat (*Eriobotrya japonica*)								
Rosaceae	Wild plum (*Prunus subcordata*)								
Loquat	Cactus (*Optunia streptacantha*)								
Rosaceae	Bitter cherry (*Prunus emarginata*)								
Rhamnaceae	Buckthorn (*Frangula purshiana*)								

Dark grey cell denotes ripening time of the fruits when they were sampled. Plant families and species were checked through the CalFlora website [65].

**Table 2 insects-10-00206-t002:** *Drosophila suzukii* emerged from fruits in the San Joaquin Valley, CA, USA.

Fruit	Collection Years	Months Sampled	Intact Fruit		Damaged Fruit	
			No. of Fruit Collected	No. of *D. suzukii* Emerged	No. of Fruit Collected	No. of *D. suzukii* Emerged
Cherry	2013–2015	May–Jun	582	372	1372	366
Peach	2013–2014	Jun–Aug	50	0	41	11
Nectarine	2013–2014	Jun–Jul	50	0	40	2
Plum	2013–2014	Jul–Aug	50	0	60	5
Pear	2013–2014	Jul–Sept	30	0	23	104
Fig	2013–2014	Sept–Nov	50	0	119	115
Apple	2013–2014	Sept–Oct	50	0	106	4
Pomegranate	2013–2014	Sept–Nov	-	-	100	12
Raisin grape	2013–2014	Oct–Nov	1020	0	20	0
Persimmon	2013–2014	Oct–Dec	50	0	15	1
Mandarin	2013–2014	Oct–Apr	-	-	55	2
Naval orange	2013–2014	Oct–Apr	-	-	20	1
Loquat	2014	May–Jun	50	0	15	5
Wild plum	2014	Jul	50	0	-	-
Cherry plum	2014	Jul	50	0	-	-
Cactus	2014	Sept	30	0	23	1
Buckthorn	2016	Oct	150	0	-	-
Bitter cherry	2016–2017	Sept–Oct	4,800	0	-	-

**Table 3 insects-10-00206-t003:** Suitability of major crop and non-crop fruits as alternative ovipositional and/or reproductive hosts for *D. suzukii.*

Fruit	Fruit Firmness (gmm^−1^) ^2^	Brix^2^	*n*	Fruit Size (g) ^2^	Eggs per Gram Fruit ^2^	Eggs Developed to Adults (%) ^2^
Cherry ‘Bing’ ^1^	76.8 ± 7.8 (15) c	21.5 ± 0.4 (10) d	25	8.56 ± 0.17 a	1.22 ± 0.18 c	63.4 ± 6.3 a
Pomegranate seed	45.8 ± 3.2 (20) d	17.0 ± 0.2 (20) b	49	0.45 ± 0.01 g	4.11 ± 0.26 b	68.7 ± 5.3 a
Mandarin orange	-	10.6 ± 0.4 (20) de	53	2.37 ± 0.08 e	1.80 ± 0.19 c	51.2 ± 5.5 ab
Raisin grape	84.8 ± 6.7 (20) c	17.9 ± 0.2 (20) c	52	1.76 ± 0.05 f	2.17 ± 0.25 c	26.5 ± 4.5 c
Table grape	181.5 ± 10.0 (20) b	21.2 ± 0.3 (20) c	33	3.50 ± 0.09 d	1.12 ± 0.25 c	31.4 ± 6.1 c
Wine grape	102.3 ± 6.3 (15) c	23.6 ± 0.5 (20) b	22	0.70 ± 0.03 g	4.84 ± 1.17 ab	4.5 ± 3.1 d
Wild plum	295.4 ± 7.9 (15) a	10.5 ± 0.2 (10) e	25	9.71 ± 0.11 a	1.10 ± 0.16 c	53.1 ± 6.6 ab
Cactus	-	11.5 ± 0.1 (20) a	22	7.21 ± 0.45 c	1.56 ± 0.12 c	64.1 ± 4.9 a
Buckthorn	-	-	53	0.69 ± 0.02 g	5.10 ± 0.39 ab	68.8 ± 5.4 a
Bitter cherry	-	-	51	0.32 ± 0.01 g	6.46 ± 0.46 a	33.2 ± 5.0 c

^1^ The cherry was the same Cultivar 1 as shown in Table 4 and here served as a companion to alternative hosts. ^2^ Values are mean ± SE. Values in parentheses refer to replicates. Different letters within a column indicate significant difference among treatments (One-way ANOVA and Tukey’s HSD, *p* < 0.05).

**Table 4 insects-10-00206-t004:** Suitability of different cherry cultivars for the oviposition and development of *D. suzukii*.

Cultivar ^1^	Fruit Color	Fruit Size (g) (*n* = 10) ^2^	Brix (*n* = 10) ^2^	Fruit Firmness (g mm^−1^) (*n* = 15) ^2^	Eggs per Gram Fruit (*n*= 25) ^2^	Eggs Developed to Adults (%) ^2^ (*n*= 25)
Cultivar 1	Purple	8.7 ± 0.4 a	21.5 ± 0.6 c	76.9 ± 7.9 d	1.23 ± 0.18 a	63.4 ± 6.3 abc
Cultivar 2	Purple	7.4 ± 0.4 b	24.0 ± 0.8 bc	99.6 ± 3.0 c	1.19 ± 0.17 a	80.3 ± 4.9 ab
Cultivar 3	Pink	6.5 ± 0.1 bc	24.6 ± 0.3 ab	99.7 ± 3.9 c	1.36 ± 0.19 a	69.1 ± 6.4 abc
Cultivar 4	Pink	4.4 ± 0.2 e	27.3 ± 0.5 a	128.7 ± 4.2 b	2.71 ± 0.55 a	53.1 ± 6.0 abc
Cultivar 5	Red	4.3 ± 0.1 e	22.6 ± 0.7 bc	156.8 ± 3.9 a	2.18 ± 0.31 a	68.4 ± 4.6 abc
Cultivar 6	Yellow	2.5 ± 0.1f	16.0 ± 0.9 d	114.4 ± 3.5 bc	4.44 ± 0.69 b	52.9 ± 5.7 c
Cultivar 7	Yellow	5.6 ± 0.3 cd	18.3 ± 0.9 d	97.6 ± 3.5 c	1.88 ± 0.26 a	77.8 ± 4.8 abc
Cultivar 8	Yellow	4.8 ± 0.2 de	22.5 ± 0.4 bc	132.8 ± 2.4 b	1.98 ± 0.29 a	61.0 ± 6.7 abc
Cultivar 9	Black	3.6 ± 0.3 ef	15.6 ± 0.8 d	100.8 ± 4.0 c	2.03 ± 0.30 a	80.2 ± 4.6 a
Cultivar 10	Black	4.4 ± 0.1 e	24.2 ± 0.6 bc	52.0 ± 4.1 e	2.35 ± 0.27 a	57.1 ± 5.6 abc

^1^ Cultivars were selected from a seeding selection trail, and the varieties have not been released yet. ^2^ Values are mean ± SE. Figures in brackets refer to replicates. Different letters within the column indicate significant difference among treatments (One-way ANOVA and Tukey’s HSD, *p* < 0.05).

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
