# Peer review of "Temporal Dynamics of Host Use by Drosophila suzukii in California’s San Joaquin Valley: Implications for Area-Wide Pest Management"

_insects, 2019, doi:10.3390/insects10070206_

Round 1

Reviewer 1 Report

This paper reports host use of a fruit pest D. suzukii in San Joaquin Valley based on field studies and laboratory experiments and provides some implications for managements this pest. The paper is well written, and I think it is acceptable with minor revision. My questions and comments are as follows.

Line 132: For adults, a 10% honey-water solution was used as food. However, honey contains little amino acids or proteins. I wonder how flies acquire nitrogen compounds. Are nitrogen compounds not necessary for adult flies (e.g., for egg production)?

Line 180: What is the cause of the occurrence of other drosophilids? Is the rearing system OK?

Lines 212-213: Currently, it is thought that analyses of proportions using arcsine transformation have some problem (e.g., Warton & Hui: Ecology 92: 3-10, 2011). I think GLM or GLMM is appropriate, as the authors performed in the analysis of egg-to-adult viability.

Line 216: As above, I think logistic regression (GLM with binomial distribution and logit-link function) is more appropriate than simple linear regression.

Lines 302-302: Please provide the statistics for the comparison of pupal body size.

Lines 322-342, 404-406: The authors note that alternative fruits may play an important role in sustaining D. suzukii populations late in season. According to the seasonal change of fly catch (Figure 4), however, the population increase in autumn in low-altitude areas was very large. I wonder whether this increase can be explained by the use of alternative fruits in summer or early autumn. In addition, although the D. suzukii infection on cherry was high in May and June (Table 1), D. suzukii catches decreased from June to July (Fig. 4). Thus, there are some discrepancies between the seasonal trends of host use and the seasonal patterns of fly catches. It is desired to provide some explanation for these discrepancies.

Lines 362-363: It is stated that the physical properties (color, size or surface firmness) of different cultivars did not seem to affect the fly’s oviposition, but the analysis on color is not given in “Results”.

Minor points

Line 34: Please amend “Matsumura” to “(Matsumura)”.

Line 61: Please amend “caneberries” to “cranberries”.

Lines 123, 210, 226, 231,336, 375 and Table 3: The word “emerge” is intransitive verb.

Line 229: Is the number of fruit (723) correct? According to Table 1, the number of intact fruit examined was 582, ant the number of damaged fruit was 1372.

Author Response

Please note that we include are comments to both reviewers as there was some overlap. As is almost always the case, the Reviewers' comment greatly improved the manuscript and we have made all of the requested changes to the current draft.

Response to reviewers’ comments:

#Reviewer 1
This paper reports host use of a fruit pest D. suzukii in San Joaquin Valley based on field studies and laboratory experiments and provides some implications for managements this pest. The paper is well written, and I think it is acceptable with minor revision. My questions and comments are as follows.

A. (A. = Authors’ responses) thank you for your careful reading of the manuscript and thoughtful and useful comments that have enhanced the overall quality of the manuscript.

Line 132: For adults, a 10% honey-water solution was used as food. However, honey contains little amino acids or proteins. I wonder how flies acquire nitrogen compounds. Are nitrogen compounds not necessary for adult flies (e.g., for egg production)?

A. We should have mentioned in the original submission that we also provided adult flies in the rearing cages with petri diches containing standard cornmeal medium sprinkled with brewer’s yeast for feeding and oviposition. The yeast was used to enhance the attraction of the diet to the adult flies. It is also possible that the flies might also feed on the yeast for protein. We have been using this method to maintain our SWD colony (so do in many other studies, although some used 20-50% honey water). It is well known that both adult males and females need intake of carbohydrates (C) for energy and longevity, but it is not clear how dietary protein (P) would enhance egg production of SWD. A recent study (Rendon et al. 2018) found that overall a low protein to carbohydrate (P:C 1:4) diet composition resulted in SWD higher reproductive values coupled with longer survival periods, while diets containing high protein (P:C 1:1) levels resulted in lower reproductive values and shorter survival periods. This study indicates the importance of carbohydrates as an essential resource for survival and suggest that dietary protein may have a crucial role in early oogenesis (i.e., accelerating early egg maturation) but at the cost of longevity.

We now have changed this sentence to “…supplied with a 10% honey-water solution and petri dishes containing standard cornmeal medium sprinkled with brewer’s yeast for feeding and oviposition”.

Line 180: What is the cause of the occurrence of other drosophilids? Is the rearing system OK?

A. In this orange test, we had a few replicates (discarded) contaminated by other drosophilids (mainly D. melanogaster). Our experiments were conducted in the same room where we also processed a lot of field-collected fruits. Although we carefully kept all samples in fine-screened cages, still a few flies might have escaped from the cages. In this experiment, we had to simulate natural decay process of a fallen orange, by placing fresh oranges individually over wet sandy soil in small cage until the fruit started to rot. We kept checking the decay status of the fruit and also measured the percentage of fruit surface covered by mold growth (42.3 ± 7.3% coverage, data were not reported originally). We believed the contamination occurred during the examination of these fruit. Our rearing system was ok, although we did have a few occasions since 2013 when we had our SWD colonies contaminated by D. melanogaster. We checked our SWD colonies (SWD pupae are obviously different from other dosophilids) all the time during the rearing to end and prevent any contaminations.

We have provided the mold information: “On average, rotted and halved rotting fruit had 42.3 ± 7.3 % of their surface covered by mold growth” and added the possible cause of contamination “… “likely occurred during the regular examination for the decay status of the fruit”.

Lines 212-213: Currently, it is thought that analyses of proportions using arcsine transformation have some problem (e.g., Warton & Hui: Ecology 92: 3-10, 2011). I think GLM or GLMM is appropriate, as the authors performed in the analysis of egg-to-adult viability.

A. We realized that the Data analyses section needs some details as both reviewers have raised some questions. We have made substantial changes. We actually used GLM for the analyses of percentage survival of SWD eggs on different cherry cultivars or different fruit species (see the original results L254-256). But we also did ANOVA for the same data in order to separate the means on Tables 3 and 4.

Line 216: As above, I think logistic regression (GLM with binomial distribution and logit-link function) is more appropriate than simple linear regression.

A. In this specific analysis, we looked at the effect of egg density in terms of eggs laid per gram fruit (using same cherry cultivar) on the percentage of eggs developed into adults. There was only one independent variable and there was a significant linear relationship between the egg density and survival. We tried to use both simple linear regression and logistic regression but found that the linear regression was a better fitting to describe the relationship. Thus, we decided to keep the linear regression.

Lines 302-302: Please provide the statistics for the comparison of pupal body size.

A. Done.

Lines 322-342, 404-406: The authors note that alternative fruits may play an important role in sustaining D. suzukii populations late in season. According to the seasonal change of fly catch (Figure 4), however, the population increase in autumn in low-altitude areas was very large. I wonder whether this increase can be explained by the use of alternative fruits in summer or early autumn. In addition, although the D. suzukii infection on cherry was high in May and June (Table 1), D. suzukii catches decreased from June to July (Fig. 4). Thus, there are some discrepancies between the seasonal trends of host use and the seasonal patterns of fly catches. It is desired to provide some explanation for these discrepancies.

A. At the low-altitude areas, temperature started raising rapidly in early June and maintained high throughout the summer and early fall (for example, the average temperature from July to September in 2013 was 31.4°C, see Wang et al. 2016).  The fly catches were very low from later June to later September when many summer fruits are ripening. Some late summer fruits such as pomegranate (cracked) and figs (cracked) were often left in the field. It is possible that they might have provided reproductive host for the fly and played a role in the resurgence of the fly population in middle fall when the temperature is suitable again. Most cherries fruit are harvested by middle June when the temperature was about to raise. Peak infestation on cherry occurred in late May (we added the specific date on the peak infestation we mentioned in the results section).  We have also made some changes to the discussion of this section.

Lines 362-363: It is stated that the physical properties (color, size or surface firmness) of different cultivars did not seem to affect the fly’s oviposition, but the analysis on color is not given in “Results”.

A. We did not include the color as a factor in the analyses because there were five different colors and each color represented only one to three cultivars. We edited this statement to not mention color.

Minor points
Line 34: Please amend “Matsumura” to “(Matsumura)”.

A. Done.

Line 61: Please amend “caneberries” to “cranberries”.

A. Done.

Lines 123, 210, 226, 231,336, 375 and Table 3: The word “emerge” is intransitive verb.

A. A. Done.

Line 229: Is the number of fruit (723) correct? According to Table 1, the number of intact fruit examined was 582, ant the number of damaged fruit was 1372.

A. Done. The number on the Table 1 included all cherries collected from 2013 to 2015. Here the number (we wanted to give an example) only included the 2013 samples with both intact and damaged fruit combined because both intact and damaged cherry fruit were attacked by SWD.
We have made changes to clarify this.

Reviewer 2 Report

General comments:

The publication is well written, the question is interesting and reasonable and the data are comprehensive. However, I see a substantial lack of clarity in the interpretation of the results. For example, some results that are announced in the material and methods chapter are not even displayed in the results section. Thus, the presentation need to be revised. Please avoid using different terms for the same thing, it makes reading much easier und better understandable (e.g. success rate, successful rate, eggs developed to adults, survival rate).

Line 21:

Please use “emerged” instead of “utilized” because it is easier to read

Line 94:

Delete: “To increase our understanding of seasonal (temporal) changes temporal” 

Lines 101-114:

It would be interesting to have more details about the sampling e.g. a table in the supplementary material where the sampled hosts per site, the number of hosts per species and in which years are given.

Why didn’t you included raspberries, blueberries, strawberries,… as they are known to be susceptible to D. suzukii?

Lines 136-159:

Please include, that the tests were no-choice tests (for clarity and to differ from the tests in the next section)

Line 167-168:

The term host density seems to used wrong here, because host would be the berry in your example and as I understand right you are talking about pest or herbivore density (and you expected from table 3 and 4 that emerging rates of the fly are negatively related to egg/larvae density?)

Lines 167-170:

Where are the results from this experiment????

Lines 213-218:

Unclear formulation: how did you calculate the success rates (egg developed to adults)? The success rates has to be calculated as number of emerged adults/number of eggs without standardizing the number of eggs to gram fruit.

Line 226: 

Delete the sentence: “No flies emerged from undamaged fruits except cherries” or rewrite the first three sentences of the section.

Lines 226-228:

Were there only pears in Brentwood or also in Fresno, were pears damaged at every sampling site or just a few, were the damaged pears infested in every year or just one? Please give a more detailed table in the supplementary material as already stated above.

Lines 228-229:

Where are these values come from? (from different years, sites, sampling sites)

Lines 234-243:

The study aimed at evaluating “the suitability of key fruits as ovipositional and or/breeding hosts for D. suzukii”. To investigate that topic it would in in first instance interesting, how oviposition rate and emergence rate differ between hosts. Please do not use standardized egg data, it is not necessary, as you used only one female per treatment, which is very limited in oviposition, thus, fruit size is not crucial and I suspect that the differences in egg density you produced on this way is just an artefact. Instead, give eggs per treatment and emerged adults per treatment. After these primary patterns are reported, you can consider the role of fruit size. Egg density effects should be investigated in a separate experiment (MM 2.2.3 fruit size preference and host density effects…).

Figure 1:

What is the percentage of egg allocation (C)? It should be possible to understand from the figure legend, but it is not even explained in Material and Methods

Lines 274-275:

I suppose, this is the result of the “host density (herbivore density)”-test? The data should be displayed (figure).

Line 277:

D. suzukiihas to be set to italic

Lines 302-303:

Statistically tested? Please include in Fig. 3.

Lines 305:

Labelling is wrong (C instead of B)

Line 344:

Please delete “and host density”

Author Response

Please note that we include are comments to both reviewers as there was some overlap. As is almost always the case, the Reviewers' comment greatly improved the manuscript and we have made all of the requested changes to the current draft.

#Reviewer 2

The publication is well written, the question is interesting and reasonable and the data are comprehensive. However, I see a substantial lack of clarity in the interpretation of the results. For example, some results that are announced in the material and methods chapter are not even displayed in the results section. Thus, the presentation need to be revised. Please avoid using different terms for the same thing, it makes reading much easier und better understandable (e.g. success rate, successful rate, eggs developed to adults, survival rate).

A. Thank you for your careful reading of the manuscript and constructive criticisms which helped us to improve the overall quality of the manuscript.

Line 21: Please use “emerged” instead of “utilized” because it is easier to read

A. Done.

Line 94: Delete: “To increase our understanding of seasonal (temporal) changes temporal”

A. Done.

Lines 101-114: It would be interesting to have more details about the sampling e.g. a table in the supplementary material where the sampled hosts per site, the number of hosts per species and in which years are given. Why didn’t you included raspberries, blueberries, strawberries,… as they are known to be susceptible to D. suzukii?

A. As suggested, we provide detailed information in a supplement table.

Yes, we could have done the samples also with these three fruits (raspberries, blueberries and strawberries).  As we mentioned in the introduction, all these three fruits are known to be infested by SWD in California. There are planted mainly in northern and costal California, although they also are increasingly being planted in the San Joaquin Valley as we mentioned in the end of the discussion section. We conducted the samples mainly at the UC Kearney Research Farm where these fruits were not available at the time of sampling.   

Lines 136-159: Please include, that the tests were no-choice tests (for clarity and to differ from the tests in the next section)

A. Done.

Line 167-168: The term host density seems to used wrong here, because host would be the berry in your example and as I understand right you are talking about pest or herbivore density (and you expected from table 3 and 4 that emerging rates of the fly are negatively related to egg/larvae density?)

A. The review is right. It should be “the number of D. suzukii egg”.

Lines 167-170: Where are the results from this experiment????

A. The results were presented originally at Lines 271-273 together with the fruit size preference test. They are now present separately following the reviewer’s suggestion (see our response below to Lines 234-243).

Lines 213-218: Unclear formulation: how did you calculate the success rates (egg developed to adults)? The success rates has to be calculated as number of emerged adults/number of eggs without standardizing the number of eggs to gram fruit.

A. The reviewer is right. We referred to the percentage of eggs successfully developed into adults.

Line 226: Delete the sentence: “No flies emerged from undamaged fruits except cherries” or rewrite the first three sentences of the section.

A. Done (deleted this sentence).

Lines 226-228: Were there only pears in Brentwood or also in Fresno, were pears damaged at every sampling site or just a few, were the damaged pears infested in every year or just one? Please give a more detailed table in the supplementary material as already stated above.

A. Pears are commercially planted mainly in Brentwood areas. Yes, we found only a few damaged fruit at each site and often only a couple of damaged pears had SWD infestation. As suggested, we have provided a supplemental table listed detailed sampling information.  

Lines 228-229: Where are these values come from? (from different years, sites, sampling sites)

A. The number on the Table 1 included all cherries collected from 2013 to 2015. Here the number (we wanted to give an example) only included the 2013 samples with both intact and damaged fruit combined because both intact and damaged cherry fruit were attacked by SWD.

We have made changes to clarify this.

Lines 234-243: The study aimed at evaluating “the suitability of key fruits as ovipositional and or/breeding hosts for D. suzukii”. To investigate that topic it would in in first instance interesting, how oviposition rate and emergence rate differ between hosts. Please do not use standardized egg data, it is not necessary, as you used only one female per treatment, which is very limited in oviposition, thus, fruit size is not crucial and I suspect that the differences in egg density you produced on this way is just an artefact. Instead, give eggs per treatment and emerged adults per treatment. After these primary patterns are reported, you can consider the role of fruit size. Egg density effects should be investigated in a separate experiment (MM 2.2.3 fruit size preference and host density effects…).

A. We first added one column on Table 2 and Table 3 to show the original data based on eggs laid per testing fruit unit, but this didn’t change the results and spread Tables 2 and 3 too wide, so we went back to the original but noted in the text the significant differences among “eggs laid per fruit unit”. We also note most of these tested fruits were not naturally attacked by SWD. Thus,  the major purpose of this test was to compare the host suitability for the fly development for which we think it was necessary to standardize the egg density per gram unit because the difference in fruit chemical property.

As suggested, we now present the fruit size preference and egg density effect experiments separately.

Figure 1: What is the percentage of egg allocation (C)? It should be possible to understand from the figure legend, but it is not even explained in Material and Methods

A. Here we refer to the percentage of total allocated to the differently sized fruit. We have made this clear both in the M&M and Results sections.

Lines 274-275: I suppose, this is the result of the “host density (herbivore density)”-test? The data should be displayed (figure).

A. Yes, we have added a figure (named Figure 2) to show the data and modified the text accordingly. Please all other figure numbers must be changed as well.

Line 277: D. suzukii has to be set to italic

A. Done.

Lines 302-303: Statistically tested? Please include in Fig. 3.

A. Done.

Lines 305: Labelling is wrong (C instead of B)

A. Done.

Line 344: Please delete “and host density”

A. Done.

Round 2

Reviewer 2 Report

Thank your for your careful revision.

Author Response

We greatly appreciate the Reviewers' and Editors comments. We have made every suggested change, as well as finding a few typos that were corrected as well. We have uploaded the revised manuscript in MS Word Tract Changes, and made replied to all comments on the file. Again, we appreciate the Editor's and Reviewers' comments. Sincerely, Kent Daane